# Factors Affecting the Compressive Strength of Geopolymers: A Review

Hengels Castillo [1,2,*], Humberto Collado [1,2], Thomas Droguett [3,4], Sebastián Sánchez [3,5], Mario Vesely [1], Pamela Garrido [3] and Sergio Palma [2]

1    JRI Ingeniería, Santiago 7770445, Chile; hcollado@jri.cl (H.C.); mvesely@jri.cl (M.V.)
2    Complex Fluids Laboratory, Department of Metallurgical Engineering and Materials, Universidad Técnica Federico Santa María, Santiago 8940572, Chile; sergio.palma@usm.cl
3    CIMS–JRI, Santiago 7850000, Chile; thomas.droguett@cimsjri.cl (T.D.); sebastian.sanchez@jri.cl (S.S.); pgarrido@cimsjri.cl (P.G.)
4    Department of Metallurgical Engineering, Universidad de Santiago de Chile, Santiago 9170022, Chile
5    Escuela de Ingeniería Química, Pontificia Universidad Católica de Valparaíso, Valparaíso 2340025, Chile
*    Correspondence: hcastillo@jri.cl

**Abstract:** Geopolymers are created by mixing a source of aluminosilicates, which can be natural or by-products from other industries, with an alkaline solution. These materials based on by-products from other industries have proven to be a less polluting alternative for concrete production than ordinary Portland cement (OPC). Geopolymers offer many advantages over OPC, such as excellent mechanical strength, increased durability, thermal resistance, and excellent stability in acidic and alkaline environments. Within these properties, mechanical strength, more specifically compressive strength, is the most important property for analyzing geopolymers as a construction material. For this reason, this study compiled information on the different variables that affect the compressive strength of geopolymers, such as Si/Al ratio, curing temperature and time, type and concentration of alkaline activator, water content, and the effect of impurities. From the information collected, it can be mentioned that geopolymers with Si/Al ratios between 1.5 and 2.0 obtained the highest compressive strengths for the different cases. On the other hand, high moderate temperatures (between 80 and 90 °C) induced higher compressive strengths in geopolymers, because the temperature favors the geopolymerization process. Moreover, longer curing times helped to obtain higher compressive strengths for all the cases analyzed. Furthermore, it was found that the most common practice is the use of sodium hydroxide combined with sodium silicate to obtain geopolymers with good mechanical strength, where the optimum SS/NaOH ratio depends on the source of aluminosilicates to be used. Generally speaking, it was observed that higher water contents lead to a decrease in compressive strength. The presence of calcium was found to be favorable in controlled proportions as it increases the compressive strength of geopolymers, on the other hand, impurities such as heavy metals have a negative effect on the compressive strength of geopolymers.

**Keywords:** Si/Al ratio; curing; impurities; water/solids ratio; compressive strength

## 1. Introduction

A geopolymer is an inorganic synthetic polymer generated through the reaction between aluminosilicate materials and alkaline agents, where after curing a semi-crystalline amorphous material is generated [1]. The curing reaction can occur both at high temperatures and at room temperature, depending on the composition of the geopolymer [2].

There are a wide variety of aluminosilicate reagents that can be used to produce geopolymers. The most common sources of aluminosilicates used for the production of geopolymers are metakaolin and by-products from other industries such as fly ash, mine tailings, red mud, slags, etc. [3–10]. There are also studies on geopolymers based on volcanic ashes [11].

Geopolymer precursor materials, both in natural as well as by-product forms, are required to be rich in alumina ($Al_2O_3$) and silica ($SiO_2$) content, preferentially in reactive amorphous form [3]. The role of these compounds is to impart the strength and setting property to the cement [12]. A concern related to aluminosilicate dissolution is the rate at which it occurs and how much of the total amorphous aluminosilicate material is available for geopolymerization [13].

In addition to the aluminosilicate reagent, an alkaline activator is needed to produce the geopolymer. The alkaline activator causes the dissolution of the raw materials [14]. It must be carefully selected because its composition has different impacts on the properties of fresh geopolymer paste and development of the mechanical strength in the hardened geopolymers [15]. The most common are alkali hydroxide and silicate solutions.

In the activation of the aluminosilicate source with NaOH (most commonly used alkaline reagent), the reaction starts with the dissolution of Al and Si, which are precursor particles in the alkaline solution, and then, followed by polymerization in the aluminum-rich first gel phase, which will be transformed into silicon-rich final geopolymer gel [16].

The product resulting from the reaction between the aluminosilicate source and the alkaline activator is an amorphous substance composed of solid phases of aluminosilicates armed based on connections of $SiO_4^{4-}$ and $AlO_4^{5-}$ as tetrahedra forming a 3D structure.

In short, geopolymerization is an exothermic process involving various oligomers and other structural units (three-dimensional structural units) that form macromolecular microstructures, which in turn determines the mechanical properties of geopolymers through numerous experimental and theoretical studies of reaction kinetics. Some researchers have proposed that the synthesis of geopolymers is composed of three steps in the following sequence (Figure 1): (i) the dissolution of aluminosilicate materials comprising silicate and aluminate monomers, (ii) the gel formation process involving the transformation of the active monomers into geopolymeric fragments of cross-linked aluminosilicate oligomers, and (iii) the formation of geopolymer gel through the chain reaction of crystallization and polymerization [17].

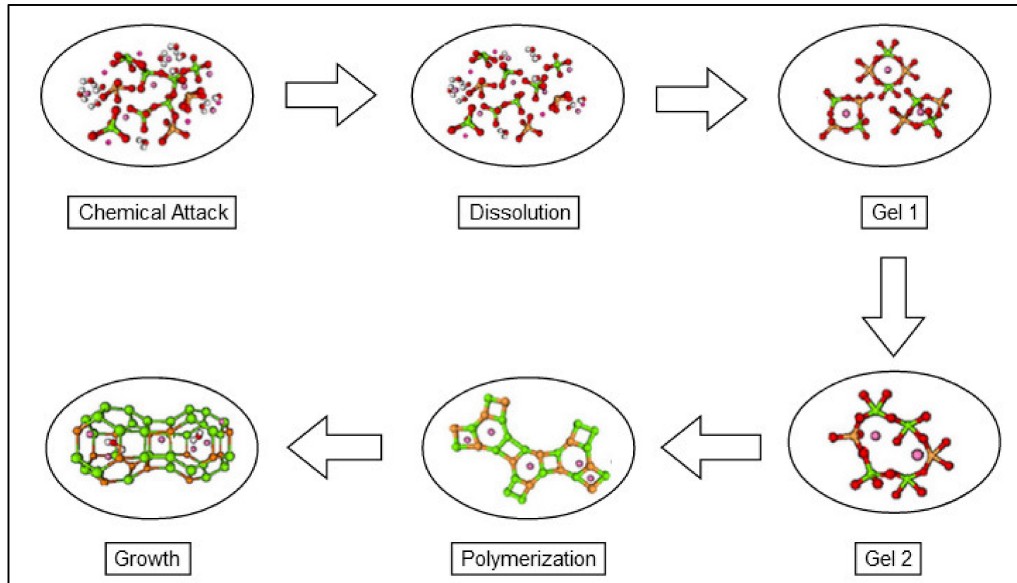

**Figure 1.** Conceptual design of the geopolymerization process.

The main hydration product of low-calcium or calcium-free binders is N-A-S-H gel, which possesses a three-dimensional structure [18]. The development of strength in geopolymers strongly depends on the raw materials and the alkali activator solutions [19].

In general, and from another point of view, geopolymers are a man-made material that offer several advantages, including good mechanical strength and the capacity to encapsulate hazardous waste, as well as being water and fire resistant.

Geopolymer is being studied extensively and shows promise as a greener alternative to Portland cement concrete [20]. Geopolymers use industrial by-products as precursors, and, therefore, result in the emission of significantly less $CO_2$ per ton of concrete produced [21]. Geopolymer concrete (GPC) is estimated to reduce 80% carbon footprint in construction projects compared with ordinary Portland cement [22,23].

In order to build a geopolymer with a good compressive strength, different variables must be taken into account at the time of designing it, such as the type of aluminosilicate source, its composition, the composition and concentration of the alkaline activator, the amount of water to be used, whether it is cured at ambient temperature or by adding heat, among others. For this reason, the most important variables were considered when designing a geopolymer to obtain a good compressive strength, which were divided into Si/Al ratio, temperature and curing time, alkaline activator, water content, and the effect of the presence of calcium and other impurities.

This paper compiles studies by several researchers, focusing mainly on the effect of the aforementioned variables on the compressive strength of geopolymers. Within the studies, different sources of aluminosilicates will be analyzed, geopolymers will be compared with classic materials such as OPC and also possible applications that have been given to geopolymers in recent years will be discussed, including their use as construction materials.

## 2. Compressive Strength

Geopolymer concrete has mechanical properties comparable to those of OPC concrete [19]. To use geopolymers as construction material, it is required that it must have good mechanical and volumetric stability [24], and especially a stable compressive strength so that the factors that have a major influence on this property will be compiled as indicated in the next chapters.

### 2.1. Molar Ratios

The early and final strength of the alkali activated materials depends on the design of the mixture (Si/Al, Al/Na, water/Na ratio, etc.) and the reactivity of the components.

Lahoti et al. [25] showed the influence of four mix design parameters (Si/Al ratio, water/solids ratios, Al/Na ratio and water/Na ratio) on compressive strength of metakaolin-based geopolymers synthesized. Figure 1 shows a clear trend, where the compressive strength increases with the Si/Al ratio, peaks at Si/Al ratio close to 2 and then decreases with increasing Si/Al ratio afterwards. On the other hand, in the study of the water/solid ratio, the dependence of the compressive strength on this ratio was analyzed, where, in general, a decrease in compressive strength was observed with increasing water content, but large variations in compressive strength were also observed with the same water/solid ratio. This indicates that the water/solid ratio alone does not determine compressive strength of alkali activated geopolymers. The variation of the Al/Na ratio showed that the highest compressive strength can be achieved when this ratio is close to one. Finally, it was observed that the water/Na ratio does not significantly influence the development of compressive strength. Further, the same author carried out an analysis to give quantitative information about the relative importance of the variables presented above. The results showed that Si/Al ratio is the most important parameter followed by Al/Na, $H_2O/Na_2O$ and water/solid ratios. The Si/Al ratio of the alkali activated materials mixtures cannot go below 1.0 as this is the lower limit for any geopolymer gel because the formation of Al-O-Al bonds is not favored [26]; however, there must be enough aluminum to have a stable aluminosilicate network, otherwise the dissolution of the excess silica will occur [27].

The study by Duxson et al. [28], in which the composition and microstructure of metakaolin-based geopolymers were studied, obtained a simple compressive strength of 75 MPa with a Si/Al ratio = 1.90 using sodium silicate ($Na_2SiO_3$) as an alkaline solution with

a Na/Al ratio = 1 and a given amount of water by the ratio $H_2O/Na_2O = 11$. Figure 1 shows that Si/Al ratios above 1.9 give a lower compressive strength. This reduction may be related to the effect of unreacted materials present, since geopolymers are known to have unreacted amounts of the aluminosilicate source [29], but this depends on the type of aluminosilicate source used. Duxson et al. [30] carried out tests varying the type of alkali used (sodium and potassium hydroxide), each in pure form or as mixtures (Table 1), in addition to the Si/Al molar ratio after 7 and 28 days of aging at ambient conditions to observe the development of the compressive strength and Young's modulus over time. The comparison between the types of alkalis used showed that there is not a great variation in the mechanical properties after seven days. However, after 28 days, the compressive strength of the sample with a mixture of the two types of alkali (where potassium represents at least half of the alkaline solution) showed an increase past the molar ratio of Si/Al = 1.90 compared to the cases in which pure alkalis are used, which show a decrease in compressive strength. This indicates that a mixed use of alkalis would allow higher Si/Al molar ratios. It should be noted that at a ratio of 2.15, the geopolymer generated with an alkaline Na50 solution achieved a simple compressive strength of 90 MPa, being the highest value compared to the results obtained with pure alkaline solutions of Na and K (60 and 50 MPa, respectively).

**Table 1.** Nomenclature for the composition of alkali mixtures in alkaline solution.

| TAG | Na% | K% |
| --- | --- | --- |
| Na25 | 25 | 75 |
| Na50 | 50 | 50 |
| Na75 | 75 | 25 |

Wan et al. [31] conducted a study of metakaolin-based geopolymers in a varied range of Si/Al molar ratios. From the obtained results presented in Figure 1, it was observed that it reached the maximum compressive strength at a Si/Al = 1.67 ratio with a compressive strength of 36.8 MPa. The Na/Al and $H_2O/Na_2O$ ratios are similar to those used by Duxson et al. [27,29] (ratios of 1 and 12, respectively), showing that a continuous increase in the Si/Al ratio implies a constant decrease in the compressive strength, where at the Si/Al ratios of 3, 4, and 5, a simple compressive strength of approximately 15, 11, and 5 MPa were obtained, respectively.

Perera et al. [32] and Steveson and Sagoe-Crentsil [33] obtained compressive strengths of 61 and 48 MPa, respectively, with a Si/Al molar ratio of approximately 2. However, in Steveson's case, the same strength value is reached with two different ratios (1.75 and 1.90), because the ratio of 1.90 was achieved by adding more sodium silicate to the solution compared to the geopolymer that was designed. Although, with a ratio of 1.75, this result shows that increasing the silicon content in the mixture by means of sodium silicate does not alter the compressive strength.

Rodríguez et al. [34] and Subaer [35] obtained optimal compressive strength with a Si/Al ratio of 1.5. Figure 1 shows the compressive strength results obtained by Subaer using a Na/Al ratio = 1. From the study, it was observed that this Na/Al ratio allows to increase the Si/Al ratio without any loss in compressive strength, as compared to lower Na/Al ratios, which greatly decrease their strength with an increase in the Si/Al ratio over 1.75. Along with the compressive strength results, Subaer studied the effect of varying the Na/Al ratio on the apparent porosity of the geopolymers and compared these results with the compressive strength results. He observed that there is an inversely proportional relationship between the two results, implying that porosity controls compressive strength to some extent, because the higher the porosity, the lower the compressive strength. It should also be noted that the permeability of materials is largely controlled by porosity (Angelone, et al. [36]), indicating that conditions that reduce compressive strength are those that subsequently increase the permeability of the geopolymer. As expected, a decrease in porosity implies a more homogeneous and compact microstructure, which in turn leads to lower permeability and higher compressive strengths [37], as shown by a study by

Sun et al. [38]. In order to obtain lower porosities in geopolymers, the curing temperature must be increased, as shown in several previous studies [39]. The pore size of alkali activated materials can be affected due to carbonation, where this increases the size of the pores, which in turn leads to losses in strength [40].

In order to obtain higher compressive strengths, several authors have added different aggregates to geopolymeric mixtures [41], such as sand [42,43], granite [44], gravel, sawdust [45], dolomite [46], glass [47–51], recycled materials [52], among others. On the other hand, it has been shown that milling of the blend of raw materials produced hydraulic cements with improved compressive strength when compared with separately milled raw materials that were blended after milling [53]. Similarly, it has been observed that grinding the aluminosilicate source to obtain a finer material also produces improvements in the compressive strength of the geopolymers produced [54–58]. The reactivity of raw materials during alkali activation is an important factor for geopolymer applications. For example, mine tailings have low reactivity, which leads to products with poor mechanical strength [59], incorporating additives like metakaolin or slag can improve the properties of the resultant alkali activated material [59–61].

Zhang et al. [62] conducted a study in which the long-term compressive strength of heat cured fly ash geopolymer concrete was analyzed. Figure 1 shows the compressive strength obtained after 480 days of aging of a fly ash geopolymer concrete activated with a 14 M NaOH solution at different Si/Al ratios. Compressive strength was first observed to increase with the increase in Si/Al ratio and with Si/Al ratio close to 1.87 tended to show high compressive strength. The compressive strength of fly ash geopolymer concretes prepared with NaOH solutions of different molarity was also analyzed in this study. To some extent, the influence of Si/Al molar ratio on the long-term compressive strength was affected by the concentration of NaOH solution.

Rodríguez et al. [34] presented a similar phenomenon in its results with respect to Subaer [35], as greater strength was obtained at a molar ratio of Si/Al = 1.5 and from then on, its values of compressive strength decreased (Figure 1). Among the results obtained in this study, it was also found that the increase in the amount of Na caused a decrease in the compressive strength of geopolymer concretes.

Riahi et al. [63] achieved maximum compressive strength at a Si/Al molar ratio of around 1.63 and a Na/Al ratio of 1, showing a bell effect like most of the investigated cases [27,29,30,34], because the compressive strength first increases with the Si/Al ratio and then decreases as the Si/Al ratio continues to increase (Figure 1).

Yunsheng et al. [64] achieved a simple compressive strength of 34.9 MPa with a Si/Al ratio of 2.75, an Na/Al ratio of 1, and an $H_2O/Na_2O$ ratio of 7. Rowles and O'Connor [65,66] obtained similar results in two studies (Table 2), obtaining a compressive strength of 64 MPa with a Si/Al ratio of 2.5 and an Na/Al ratio of 1.29.

**Table 2.** Results of compressive strength at different ratios of Si/Al and Na/Al [65].

| Si/Al | Simple Compressive Strength [MPa] | | | | | |
|---|---|---|---|---|---|---|
| | Na/Al | | | | | |
| | 0.51 | 0.72 | 1.00 | 1.29 | 1.53 | 2.00 |
| 1.08 | 0.4 | 2.2 | 4.4 | - | - | - |
| 1.50 | - | 6.2 | 23.4 | - | 19.8 | - |
| 2.00 | - | - | 51.3 | 53.1 | - | 11.8 |
| 2.50 | - | - | - | 64.0 | 49 | - |
| 3.00 | - | - | - | - | 2.6 | 19.9 |

Zhang et al. [17] conducted a study based on fly ash and copper tailings from a mine in the United States, where he worked with different proportions of these two sources. The work of Zhang et al. showed that as the amount of tailings increased,

the compressive strength decreased, so the best result was with 25% tailings and 75% ashes, which gave a Si/Al ratio of 2.38. It should be noted that at the highest value of Si/Al (7.78), it subsequently obtained a compressive strength between 1 and 3 MPa when the molar concentration of the alkaline solution was varied. The constant reduction of the compressive strength when increasing the content of tailings in the formation of the geopolymer is because this alone has a very high Si/Al ratio. Therefore, increasing the proportion of tailings with respect to fly ash will imply an increase in the Si/Al ratio that ends up delivering low values of compressive strength. In addition, the amount of soluble silicon and aluminum that the tailings can contribute to the formation of the geopolymer is low because the tailings are composed mainly of crystalline phases, since they do not undergo a high temperature process in their formation.

Singh et al. [67] conducted a study based on the behavior of red mud based geopolymers cured at room temperature, with the addition of different aluminosilicate sources such as fly ash, slag, and microsilica. The study showed that an increase in the Si/Al ratio led to an increase in the compressive strength of the geopolymer concrete (Figure 1), the optimum point being at Si/Al ratio = 2 ($SiO_2/Al_2O_3 = 4$) where a compressive strength of 40 MPa was obtained. Higher amounts of silica led to a loss in compressive strength. This may be due to the small amount of aluminum in relation to the amount of silicon, so that with high silicon contents there would be unreacted silicon and, therefore, not so much geopolymeric gel would be formed [68].

The following is a compilation plot of the compressive strength data as a function of the Si/Al ratio collected from the above-mentioned investigations. In general, it can be observed that most of the authors obtained the highest compressive strengths using a Si/Al ratio between 1.5 and 2. However, it should be noted that the strength obtained depends not only on the Si/al ratio, but also on many factors such as the type of aluminosilicate source, the type of alkaline activator and its concentration, curing temperature, among others; therefore, the strength obtained cannot be attributed only to the silicon and aluminum content.

Table 3 shows the ratios obtained for the different researchers to achieve the compressive strength using geopolymers. From the studies analyzed, the Si/Al ratio varied from 1.5 to 2.75, being in most cases, values close to 2 where the highest compressive strengths were obtained. The Na/Al ratio varied from 0.6 to 1.3, with the most commonly used ratio being 1.0. The $H_2O/Na_2O$ ratio of 11 was the most used by the majority of the authors. It should be noted that the resistance obtained also depends on other variables such as the source of aluminosilicates and the way of preparation of the geopolymeric mixtures.

Duxson et al. [28] studied the microstructure of alkali activated materials to find out how the Si/Al ratio influenced it. It was concluded that the N-A-S-H gel of the geopolymer is what controls the compressive strength of the geopolymer. It was evident that, at low Si/Al ratios, the material would appear more porous since there would not be enough N-A-S-H gel formation to give homogeneity to the geopolymer. As the Si/Al ratio is increased, a greater homogeneity of the mixture is observed and, therefore, a reduction in porosity. This shows that the same variables that improve the strength of the geopolymers are the same that allow the material to be less porous and, accordingly, less permeable.

The study shows that there is an abrupt change in the microstructure when the Si/Al ratio increases from 1.45 to 1.60, qualitatively showing that the porosity is significantly reduced to accommodate a more homogeneous material.

**Table 3.** Summary of bibliographic research on the influence of ratios on mechanical strength (CT = Copper tailings).

| Source | Ratio Si/Al Optimal | Na/Al Ratio | H$_2$O/Na$_2$O Ratio | Mixing and Setting Conditions | UCS Strength [MPa] | Reference |
|---|---|---|---|---|---|---|
| Metakaolin | 1.9 | 1 | 11 (H$_2$O/Na$_2$O) | 10 min of mechanical mixing. Vibration for air removal. Cured at 25–30 °C for 24 h. | 81.6 | Lahoti et al. [25] |
| Metakaolin | 1.9 | 1 | 11 (H$_2$O/Na$_2$O) | 15 min of mechanical mixing. 15 min of vibration. Cured at 40 °C for 20 h. | 75 | Duxson et al. [28] |
| Metakaolin | 1.9 | 0.75 (K/Al = 0.25) | 11 (H$_2$O/Na$_2$O) | 15 min of mechanical mixing. 15 min of vibration. Cured at 40 °C for 20 h. 28 days of rest at ambient conditions. | ~95 | Duxson et al. [30] |
| Metakaolin | 2 | 1 | 12 (H$_2$O/Na$_2$O) −1.12 g/mL (solid/liquid ratio) | 5 min of mechanical mixing. 3 min vibration. First cures at 60 °C for 6 h and then at room temperature for 7 days. | 36.8 | Wan et al. [31] |
| Metakaolin | 2 | 1 | 7.2 (H$_2$O/Na$_2$O) | Mixed for 5 min. Vibration for 5 min. Room temperature cure for 24 h, then cure at 40 °C for 24 h. | 61 | Perera et al. [32] |
| Metakaolin | 1.75–1.9 | 1.2 | 12 (H$_2$O/Na$_2$O) | Cured at 85 °C for 2 h. | 48 | Steveson et al. [33] |
| Metakaolin | 1.5 | 0.6 | 10 (H$_2$O/Na$_2$O) | Mixed for 5–10 min. Vibration for 2 min. Cured at 70 °C for 2 h. | 86 | Subaer [35] |
| Fly ash | 1.87 | 1.2 | 11 (H$_2$O/Na$_2$O) | Mixed for 8 min. Vibration for air removal. Cured at 80 °C for 24 h. | 88 | Zhang et al. [62] |
| Metakaolin | 1.5 | 0.75 | 12 (H$_2$O/Na$_2$O) | Mixed for 12 min. Vibration for 5 min. Rest in airtight container for 7 days with relative humidity of 90%. | 35 | Rodríguez et al. [34] |
| Metakaolin | 1.63 | 0.9 | 11.25 (H$_2$O/Na$_2$O) | Mixed for 10 min. Vibration for 2 min. Cured at 50 °C and 90% relative humidity for 24 h. | ~60 | Riahi et al. [63] |
| Metakaolin | 2.75 | 1 | 7 (H$_2$O/Na$_2$O) | Mixed for 3 min. Vibration for 2 min. Cured at 20 °C and 95% relative humidity for 28 days. | 34.9 | Yunsheng et al. [64] |
| Metakaolin | 2.5 | 1.3 | 15 moles of water per 1 of metakaolin | Cured at 75 °C for 24 h. | 64 | Rowles et al. [65] |
| Metakaolin | 2.5 | 1.25 | 111 g of H$_2$O per 100 g of metakaolin | Cured at 75 °C for 24 h. | 65 | Rowles et al. [66] |
| Copper tailings and fly ash | 2.38 (25% CT) | 0.94 (25% CT) | 27% (water/solids) | Mixed for 10 min. Vibration for 2 min. 7 days of curing at 60 °C. | 14 (25% CT) | Zhang et al. [17] |
| Red mud and fly ash | 2.45 | 0.8 | 30% (water/solids) | Mixed for 5 min. Cured at 60 °C for 24 h. | 38 | Singh et al. [67] |
| Gold mine tailings | 10.7 | 0.04 | 26% (water/solids) | Mixed for 15 min and molded. Cured at 80 °C for 5 days. | 10 | Falayi 2019 [69] |
| Garnet tailings and metakaolin | 6.6 | 0.04 | - | Mixed for 10 min. Vibration for 5 min. Cures at 40 °C for 3 days. | 46 | Wang et al. [70] |
| Iron ore mine tailings | 5.98 | - | - | Mixed for 10 min. Cured at 80 °C for 3 days. | 34 | Kuranchie et al. [71] |
| Coal gangue, blast furnace slag and lead-zinc tailings | 2.0 | - | 27% (water/solids) | Mixed and vibrated for 5 min. Cured at 30 °C. | 91.13 | Zhao et al. [72] |

It was observed that at Si/Al ratios of 1:1, a small amount of N-A-S-H gel is formed, and a certain amount of zeolite nuclei is present. However, most of these are not dispersed in the binder, causing the formation of macropores. At Si/Al ratios of 2:1, a homogeneous geopolymer is formed due to the large formation of N-A-S-H gel dispersed in the matrix. At Si/Al ratios of 3:1, derivatives of soluble silicates (e.g., silicic acid) are observed in the geopolymer, allowing the N-A-S-H gel to lose its predominance. At 4:1 ratio, the N-A-S-H gel is not observed, and many micropores are formed, thus generating a large network of interconnected pores.

A study by Lahoti et al. [73] showed that by varying the Si/Al ratio of metakaolin-based geopolymers, different properties of compressive strength and volumetric stability

are obtained after being subjected to high temperatures. From the study it was obtained that with a Si/Al ratio = 2, the highest strength endurance and the lowest volume reduction are obtained. This was mainly due to the fact that with this composition, the geopolymer matrix is denser, which resulted in higher volumetric stability.

### 2.2. Curing Temperature and Time

Curing conditions largely control the formation of alkali activated materials, as elevated temperatures increase the rate of chemical reactions [74] and dissolution of reactive species [75,76], therein increasing the interaction between the aluminosilicate source and the alkaline solution at the time of the geopolymer synthesis [77]. Previous research has shown that both curing time and curing temperature significantly influence the compressive strength of geopolymer concrete [78–80].

Tian et al. [81] analyzed the effect of curing temperature on the microstructure of a geopolymer based on Chinese copper tailings and fly ash. From the results, it was obtained that with a curing time of 48 h, the compressive strength after 3, 7, and 28 days of aging in ambient conditions does not vary much, with the optimum temperature for the three cases being 80 °C, which demonstrates the rapid geopolymerization reaction. There is a change in the compressive strength of 25 °C to 80 °C of around 25 MPa, and when going from 80 °C to 120 °C, the compressive strength decreases by around 12 MPa (Figure 2).

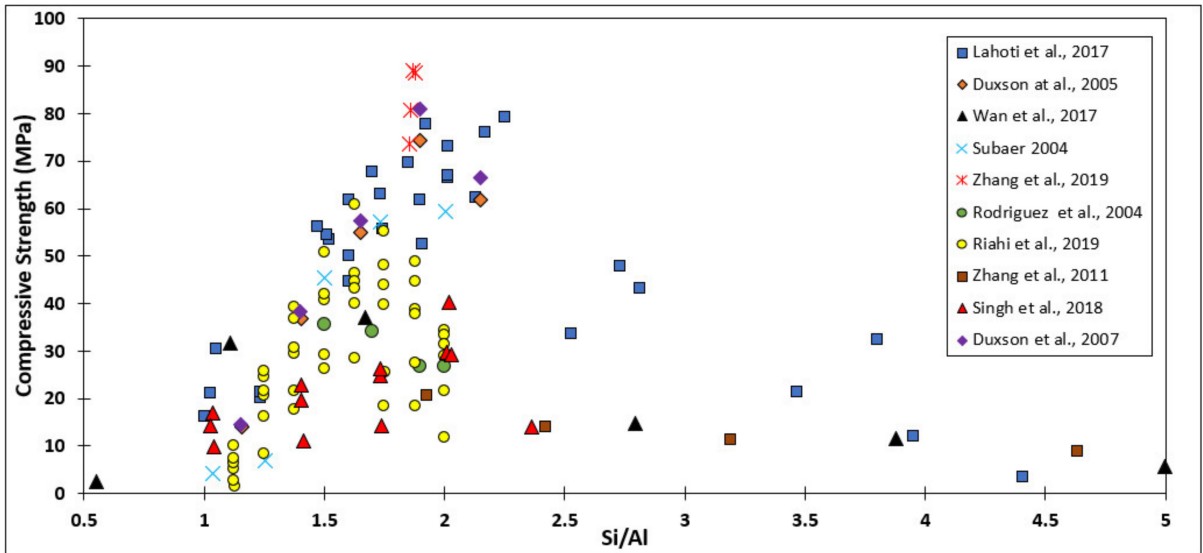

**Figure 2.** Graph compiling compressive strength vs. Si/Al ratio data obtained from previous investigations [17,25,28,30,31, 34,35,62,63,67].

Previous studies show that the conditions in which alkaline activated materials are cured must be controlled, due to the possible carbonation that can occur due to the contact between the alkaline geopolymer mixture and the $CO_2$ coming from the air [82–84]. The mechanism of carbonation in alkali-activated geopolymers is obviously different from that which takes place in Portland cement [85]. In Portland cement pastes, atmospheric $CO_2$ dissolves in the pore solution and reacts rapidly with portlandite to form $CaCO_3$, and then with Calcium–Silicate–Hydrate gel (C–S–H) to form $CaCO_3$ and silica gel [86]. In contrast, the carbonation of alkali-activated pastes occurs directly in the Calcium–Aluminosilicate–Hydrate gel (C–A–S–H) because of the lack of portlandite, leaving an alumina-containing remnant siliceous gel in addition to $CaCO_3$ [87,88].

A relative humidity of 95% in the curing chambers can inhibit the initial carbonation in geopolymers [89]. Other studies show that the optimum humidity in the curing chamber is 70%, which leads to improvements in the compressive strength of geopolymers [90].

Kong et al. [91] conducted an investigation to observe the behavior of alkali activated materials when subjected to an environment of high temperatures after their formation

(650–800 °C) and to compare their behavior with alkali activated materials without exposing them to high temperatures. In the mixture without exposure to high temperatures, the behavior was similar to that of Tian et al. [81], where the optimum curing temperature to achieve high compressive strength was at 80 °C for a period of 24 h and thereafter the value of compressive strength decreased with increasing curing temperature. Kong et al. relates this phenomenon to the evaporation of water in the pores of the geopolymer at high temperatures. However, in mixtures exposed to high temperatures, this effect is not evident, since the moisture in the system is completely dissipated when at temperatures above 650 °C, so the strength inhibiting effect is lost by evaporation of the water.

Hardjito and Rangan [92] studied the behavior of fly ash-based geopolymers, where they observed the influence of using different temperatures and curing times. Two mixtures were analyzed in the study, which are distinguished by their NaOH concentration ("Mixture 2": 8 M, and "Mixture 4": 14 M). Both mixtures were subjected to different curing temperatures, the first for 24 h and the second for 6 h. It was observed that the three experiments achieved their optimum point of compressive strength at 90 °C (similar to that presented by Tian et al. [81] and Kong et al. [91]), indeed, the mixture having better performance with 14 M NaOH. When comparing the results of "Mixture 2" at different times, it was concluded that raising the curing time from 6 h to 24 h for the same NaOH concentration achieved an increase of around 25 MPa.

Ahmari et al. [93] studied the relationship between curing temperature and alkalinity of a copper tailings geopolymer concrete based on the concentration of NaOH used. It was observed that at low alkalinity (5–10 M NaOH), the influence of the curing temperature is not so significant, whereas at a concentration of 15 M NaOH a drastic increase in strength was observed at a temperature of 90 °C. The results presented by Hardjito [92] presents a similar phenomenon, where the mixture with 15 M NaOH gave better results at a similar curing temperature. Therefore, it should be acknowledged that in a system with high alkalinity, the curing temperature is more influential and is optimal at a temperature between 80–90 °C [78,88,89]. Additionally, Ahmari et al. [93] observed the Si and Al concentration in the geopolymer at different temperatures and NaOH concentrations (Table 4) where an increase in the concentrations of both elements was observed with increasing curing temperature and alkalinity. This allows a greater dissolution of Si and Al from the aluminosilicate source, and since there is an increase in the presence of these elements in the geopolymer with increasing NaOH concentration, this implies that there is a greater contribution of these species to the formation of the NASH gel.

**Table 4.** Si and Al concentrations in the geopolymer at different curing temperatures and NaOH concentrations [93].

| Composition | Temperature (°C) | | | | | |
| --- | --- | --- | --- | --- | --- | --- |
| | 60 | | | 90 | | |
| NaOH (M) | 5 | 10 | 15 | 5 | 10 | 15 |
| Si (ppm) | 71 | 171 | 233 | 1846 | 3970 | 4570 |
| Al (ppm) | 28 | 76 | 121 | 299 | 319 | 550 |
| Si/Al | 2.44 | 2.16 | 1.85 | 5.93 | 11.9 | 7.98 |

Manjarrez et al. [94] conducted a study in which the effect of the curing temperature of tailings and low-calcium slag based geopolymers activated with a mixture of sodium silicate and sodium hydroxide (SS/NaOH = 1) was analyzed. The research shows that for a geopolymer made of 50% slag and 50% tailings, the temperature has a significant effect on the mechanical properties of geopolymer. It was shown that higher curing temperatures produce geopolymers with higher compressive strength. Due to the temperature range studied, no optimum curing temperature point was found, so it follows that

higher strengths would have been obtained if higher curing temperatures (e.g., 90 °C) had been used.

In general, it can be observed that using moderately high curing temperatures (between 80 and 90 °C) induces an improvement in the compressive strength of alkaline active geopolymers [95], because a higher temperature causes a higher interaction between the components of the mixture, higher reaction kinetics, as well as acceleration in the polycondensation process and formation of a hardened structure due to an increase in the dissolution of amorphous phases [96]. This can be seen in the compilation graph (Figure 2), which shows the compressive strength obtained at different curing temperatures by the different authors in the above-mentioned investigations. Although an optimum temperature between 80 and 90 °C was found in all the cases analyzed, the compressive strength obtained from one study to another varied greatly. This is due to the fact that the strength generated by the geopolymers does not depend simply on the curing temperature, but on more important variables such as the composition and type of aluminosilicate source.

Sun and Vollpracht [97] investigated the performance of fly ash and metakaolin-based geopolymer concretes over a one-year period, highlighting among the variables, studying the temperature and the curing time with which the geopolymer was formed and also the aging time under ambient conditions. Figure 3 shows the evolution over time of a fly ash geopolymer, where there is a significant increase during the first 100 days of aging of the geopolymer. After 100 days of aging at ambient conditions, the geopolymer does not present drastic changes in its strength. It was observed that the compressive strength does not undergo a decrease throughout the aging of the geopolymer for any of the cases.

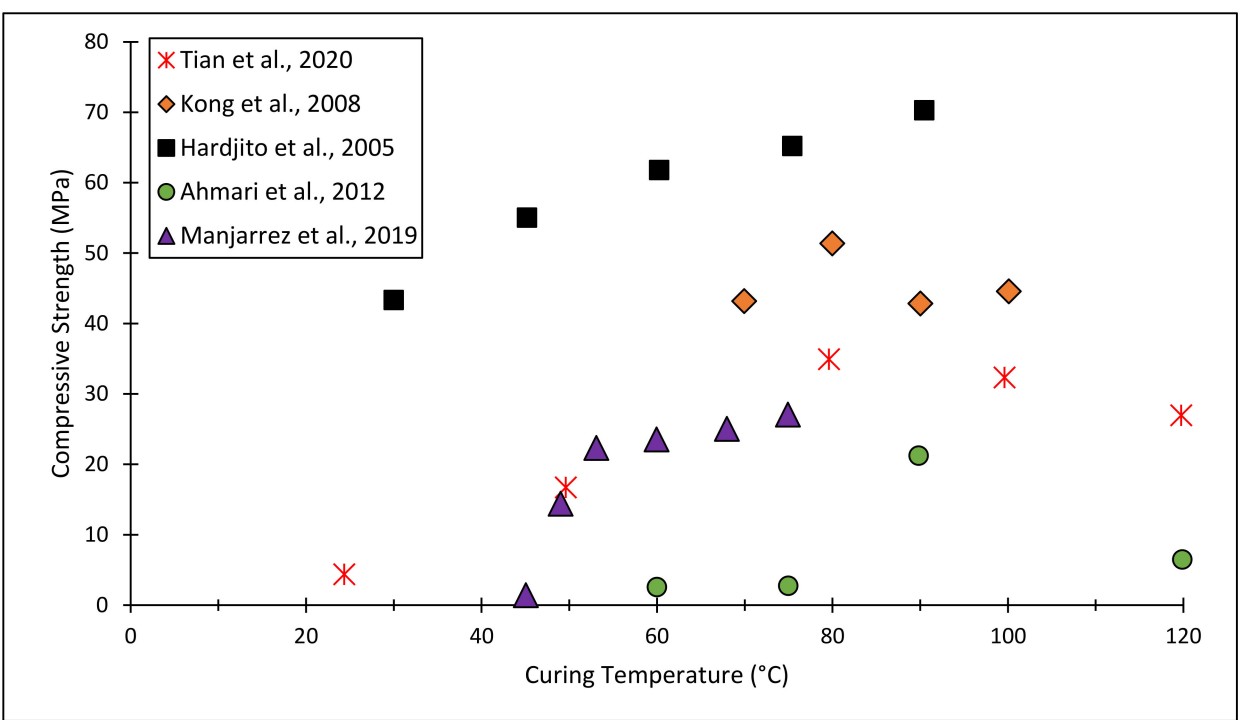

**Figure 3.** Graph compiling compressive strength vs. curing temperature data obtained from previous investigations [81,91–94].

Hardjito et al. [92] also analyzed the effect of curing time on low-calcium fly ash-based geopolymers cured at 60 °C (Figure 3). It accordingly demonstrates that the compressive strength subsequently increases with the curing time, also to be noted, increasing drastically within the first 24 h, then decreased to a slower rate, which may indicate that the dissolution of much of the aluminosilicate source occurs within the first hours of curing.

Villa et al. [98] conducted a study analyzing the behavior of alkaline-activated zeolite-based geopolymers, where he found that the highest compressive strengths were obtained

at a curing temperature of 40 °C. It is also observed that lower curing temperatures required longer curing times [99].

Okoye et al. [100] investigated the mechanical properties of alkali activated fly ash/kaolin based geopolymer concrete. Among the results of his research, he found that the highest compressive strengths were obtained with a mixture of 50% fly ash and 50% Kaolin. On the other hand, he analyzed the effect of curing time on the compressive strength of these geopolymers using different alkaline alkali activators (KOH and NaOH); from this it was obtained that for both cases, the compressive strength increased with curing time, the maximum being at 28 days of curing. In turn, the geopolymers activated with NaOH obtained higher strengths than those activated with KOH.

Samantasinghar and Prasad [101] conducted a study in which they analyzed the behavior of geopolymer concretes made with granulated blast furnace slag and fly ash in variable contents, activated with an 8 M sodium hydroxide solution. In this study, a total of six different fly ash-slag mixtures were prepared by varying fly ash and slag percentages at intervals of 20%. The investigation shows that an increase in slag content in the mixture results in an increase of compressive strength. The high reactivity of soluble alumino-silicate material in the alkaline media causes an increase of the silicon and aluminum content in the aqueous phase. These alumino-silicates make the polycondensation process more efficient, which helps in the formation of a good quality matrix. Thus, a stronger material possessing high compressive strength is obtained. Granulated blast furnace slag contains mostly reactive alumino-silicates and the leaching of ions from this slag is much higher under a given alkaline condition than from fly ash [101].

In this study, it was observed that the curing period plays a critical role in the development of compressive strength. Further, at a specified curing period, the slag rich specimen registers higher strength values than fly ash specimen. No such significant strength gain is observed with increasing cure duration for mixtures with higher slag content. The mix S100 gained about 70% of its 90 days strength at the age of seven days (Figure 3) whereas for S0, it is 11%. The rate of reaction for fly ash is relatively slow compared to slag. The presence of calcium-bearing compounds in GGBS promotes quick setting, which gives early strength [101].

In recent years, several alternatives have been studied to replace the classic cement as a building material. Geopolymers, in the family of inorganic aluminosilicate binder, has received extensive interests because of high temperature resistance [70,102–105], low permeability [106–110], strong bonding and good durability [51,111–113], excellent chemical corrosion resistance [114–118], and environmental friendliness [119,120], etc. Some of the uses that have been investigated for alkaline activated materials are: as sustainable construction material [121–126], mine backfilling [127,128], porous spheres as novel pH buffers [129,130], bricks [71,112,131,132], porous thermal insulation material [133], as road material [134], for immobilization of toxic metals and nuclear waste management [135–140], coatings for concrete [141], as waterproof surface [142], etc.

Comparative studies regarding the production price of geopolymers relative to the production price of OPC indicate that geopolymers are likely to be at a price performance disadvantage under current pricing structures. Some studies show that geopolymers range from 7% below the OPC production price to 39% higher, depending on the composition of the geopolymer [143,144]. This is why alternatives should be sought to lower the production costs of geopolymers, such as the use of waste from other industries as sources of aluminosilicates and activators.

On the other hand, it is possible to obtain hardened alkaline activated geopolymers without the need to be heated in a furnace [145], i.e., they harden at ambient temperature [57,146–149]. Some studies try to give uses to geopolymers cured at ambient temperature, as in the case of three-dimensional concrete printing [150–152].

A study by Somna et al. [54] shows the compressive strength obtained from an alkali activated material based on NaOH-activated grounded fly ash after 60 days of curing at room temperature. From the results presented in Figure 3, it can be observed

that as in heat-cured geopolymers, time plays an important role in the development of compressive strength of geopolymers cured at ambient temperature. Geopolymers with higher concentrations of the alkaline activator developed higher strengths and in a shorter time, with the optimum concentration found at 14 M.

Although some studies show that hot cured geopolymers have better compressive strengths than geopolymers cured at ambient temperature [153–155], the latter can also exhibit high compressive strengths [156,157], as shown in a study by Khan et al. [158], where a compressive strength of 108 MPa was obtained with an alkali activated material based on fly ash and slag.

Some authors have studied the effect of using seawater for the manufacture of alkaline activated geopolymers [159], where it has been observed that the use of salt water tends to improve the compressive strength of geopolymers cured at room temperature [160].

Ding et al. [161] carried out a study analyzing the properties of ambient temperature cured geopolymers based on slag and fly ash. From the results of the study, it was found that an increase in the amount of slag led to better compressive strengths. The highest strength gain was obtained in the first 28 days of curing (64 MPa), since at 90 days of curing a strength of 79 MPa was obtained, increasing only 15 MPa (Figure 3).

Previous research has shown that the hardening process (at room temperature) can be accelerated by blending FA with calcium rich source materials like granulated blast furnace slag [162].

In general, it can be observed that longer curing times produce higher compressive strengths in the geopolymers, regardless of the source of aluminosilicates (Figure 3). Further increasing the curing time did not produce a decrease in compressive strength for any of the observed cases. Although the compressive strength continues to increase with curing time, the majority of this strength was generated in the first 28 days of curing for most of the cases, with curing after this time not being as relevant in the increase of compressive strength. It should be made clear that the strength gain with curing time also depends on other factors, such as geopolymer composition and alkali activator concentration. It is also noted that the aging time at ambient temperature, after the curing time, also produces improvements in the compressive strength of the geopolymers. As in the previous graph, the huge variation observed in the compressive strength of geopolymers is mainly due to the type of aluminosilicate source used.

Table 5 shows the optimal conditions regarding temperature and curing time, together with the aging of the geopolymers prior to the tests to achieve the best compressive strength by various researchers.

**Table 5.** BiblioDiagram compilation of optimal curing conditions based on different authors.

| Source | Curing Temperature (°C) | Curing Time (1) | Aging Time (2) | Setting Time (1 + 2) | UCS (MPa) | Si/Al | Reference |
|---|---|---|---|---|---|---|---|
| Tailings and fly ash | 80 | 48 h | 28 days | 30 days | 36 | 2.84 | Tian et al. [81] |
| Metakaolin | 80 | 24 h | 3 days | 4 days | 52 | 1.54 | Ahmari et al. [93] |
| Fly ash | 90 | 24 h | 7 days | 8 days | 70 | 1.71 | Hardjito and Rangan [92] |
| Tailings | 90 | 7 days | 6 h | 174 h | 23 | 7.78 | Kong et al. [91] |
| Fly ash/Metakaolin | 20 | - | 350 days | 350 days | 70/73 | 2.93/1.81 | Sun and Vollpracht [97] |
| Fly ash/kaolin | 100 | 72 h | 28 days | 31 days | 33 | 1.47 | Okoye et al. [100]. |
| Blast furnace slag/Fly ash | 20 | - | 90 days | 90 days | 31 | 1.84 | Samantasinghar and Prasad [101]. |
| Tailings and copper slag | >75 | 7 days | 1 days | 8 days | 25 | 4.94 | Manjarrez et al. [94] |

Compressive strength changes affected by curing conditions are also related to the microstructure of the geopolymer. Tian et al. [81] concluded in their results, the different ways in which its microstructure is affected:

- At temperatures of 20 °C, cracks were observed in the geopolymer in addition to the appearance of silicon and aluminum without dispersal in the matrix. This is probably due to a poor dissolution of the aluminosilicates, which does not allow a formation of the NASH gel. At temperatures of 80 °C, homogeneity was observed in the structure of the geopolymer, indicating that silicon and aluminum are dispersed in the matrix. At temperatures of 120 °C, the distribution of silicon and aluminum continued to be observed; however, cracks reappeared and suggests the product of a decrease in the formation of the N-A-S-H gel.
- Through X-ray diffraction, they observed that there is a dissolution of the crystalline phases up to 80 °C, and it subsequently increases again when exceeding 100 °C. This would indicate that at moderately high temperatures, it is possible to provide silicon and alumina to promote gel formation. This may be due to the materials used as the source of aluminosilicates, the alkaline conditions, and the subsequent cure time.
- At temperatures above 100 °C, efflorescence is observed, where it is seen that its highest phase is $Na_2CO_3 \cdot 7H_2O$. Tian et al. indicates that this may be due to the fact that the alkaline activator (NaOH) did not have time to react completely and was exposed to the evaporated water when it was above 100 °C (consider that for the formation of the geopolymer, it was mixed for 13 min, then 6 h of curing at 100 °C and later it was left to age at ambient conditions for different periods (3, 7 or 28 days) prior to the tests carried out on the geopolymer). Moisture and carbon dioxide from the environment are absorbed and forms $Na_2CO_3 \cdot 7H_2O$, which decreases the alkalinity of the medium, and, therefore, the dissolution of aluminosilicates is reduced.

### 2.3. Alkaline Activator

As mentioned above, in addition to the aluminosilicate source, an alkaline activator is needed for the geopolymerization process to occur. The commonly used alkaline activators in the geopolymerization process are sodium hydroxide (NaOH), sodium silicate ($Na_2SiO_3$), potassium hydroxide (KOH), and potassium silicate ($K_2SiO_3$) [163,164].

The type of alkali cation is also important. It was shown that geopolymers based on a mixture of potassium silicate and KOH exhibit higher mechanical properties than those based on sodium silicate and NaOH or potassium silicate/NaOH mixtures [165,166]. The type and concentration of alkali solution affect the dissolution of the aluminosilicate source [167].

Most studies supported that the presence of alkali silicate solution in alkali reactant solution is essential and leads to better microstructure and strength properties [168]. In the reaction process, alkali silicates are combined with hydroxides to achieve better dissolution of the solid precursor and higher reaction rates [169].

The concentration of the activator has a significant effect on the compressive strengths of the geopolymers [170,171]. The ideal concentration of the activator increases the strength of the geopolymer. Moreover, an increase in the concentration of the alkaline activator leads to an increase in the pH of the activating solution. Different authors recommend working at pH values between 13 and 14 for a correct dissolution of aluminosilicates. This is corroborated by previous studies showing that higher value of hydrogen potential (pH) exhibits higher compressive strength in geopolymer concrete [172]. On the other hand, it has been shown that cement mortars with smaller particle sizes obtain higher pH values than those containing larger particle sizes [173]. Similarly, an aluminosilicate source with larger particle size tends to react less with the activating solution, due to the smaller exposed area [174].

The release rate of silicate and aluminate species from source materials is critical in controlling the synthesis process of geopolymers and the development of binding gel [175].

A high initial dissolution rate of silicate and alumina is known to accelerate the conversion of aluminosilicate materials to geopolymers [176].

It has also been observed that geopolymer mortar samples manufactured with higher alkali content perform better against attack by corrosive chemicals than those manufactured with lower alkali content [177].

Abdullah et al. [178] conducted a study in which they analyzed the effect of varying the amount of alkaline activator, as well as the effect of varying the ratio of sodium silicate to sodium hydroxide in fly ash-based geopolymers. The study did not show clear trends, but a maximum compressive strength of 70 MPa was obtained using a fly ash/alkaline activator ratio of 2.0 and a sodium silicate/NaOH ratio of 2.5, similar to that obtained in other studies [179]. In general, the compressive strength increases with the amount of fly ash and alkaline activator concentration; this is due to the increase in sodium content, which is required for the geopolymerization reaction. It is observed that at a sodium silicate/NaOH ratio of 3, the compressive strength decreases, due to the excess of $OH^-$ in the mixtures, in addition to the excess of sodium that can cause the formation of calcium carbonate when it comes in contact with the $CO_2$ in the air.

In the study carried out by Manjarrez et al. [94], where the effect of adding low-calcium slag to copper tailings-based geopolymers, in addition to varying the sodium silicate (SS)/NaOH ratio was analyzed. It was obtained that the optimum ratio of SS/NaOH was 1.0 since it obtained the highest compressive strength with all slag contents. High ratio of alkali activator/fly ash and SS/NaOH (or KOH) does not necessarily lead to high compressive strength [180]. It was also observed that at higher slag contents, the composition of the alkaline activator plays a more important role, since it produces greater increases in the compressive strength compared to when the slag is not added. On the other hand, it is clearly observed that an increase in the slag content produces an increase in the compressive strength of the geopolymer, this variable having a greater effect than the variation in the composition of the alkaline activator. The authors comment that the higher the slag content the better the fluidity of the slurry, so that a smaller amount of water is required to form the mixture; this is similar to what has been observed by other authors [181,182]. Using less water is favorable because it implies using a lower amount of alkaline reagents to reach the desired concentration, which in turn generates a benefit in the production cost of the geopolymers. The increase in compressive strength with increasing slag content is attributed to the physical and chemical properties of the slag, since the slag powder is finer than the copper tailings, so the particles have a larger surface area, which favors chemical reactions with the alkaline solution. In addition, the slag has a higher reactivity than the mine tailings due to the high temperatures to which it was subjected in the fusion process, which generates an amorphous structure ideal for geopolymerization [183].

Pavithra et al. [184] conducted a study in which they analyzed the variation of SS/NaOH ratio in fly ash based geopolymers. From the study it was obtained that the optimum SS/NaOH ratio was 1.5, obtaining a compressive strength of 46 MPa. Ratios with higher values obtained a decrease in compressive strength because higher sodium silicate contents implied an increase in the Si/Al ratio, which in turn implied a decrease in compressive strength as discussed above.

In general, no clear pattern was found for the compressive strength obtained by varying the SS/NaOH ratio (Figure 4). Some authors obtained the highest compressive strengths with SS/NaOH ratios between 1 and 1.5, while other authors found that an SS/NaOH ratio of 2.5 is the optimum value to obtain the highest compressive strengths in geopolymers (Figure 5). It should be noted that the type of aluminosilicate source is an important factor to consider, since according to the data collected, fly ash based geopolymers obtained higher compressive strengths for any SS/NaOH ratio than tailings based geopolymers, due to their low reactivity to alkaline solution. It should be noted that in most cases, a combination of sodium hydroxide and sodium silicate is used, because the latter alone is not capable of providing the necessary alkalinity to the system for a good dis-

solution of the aluminosilicates. In addition, the use of sodium silicate is preferred because some aluminosilicate sources have low reactivity, such as mine tailings, so the addition of sodium silicate helps to increase the silicon content available for geopolymerization to occur.

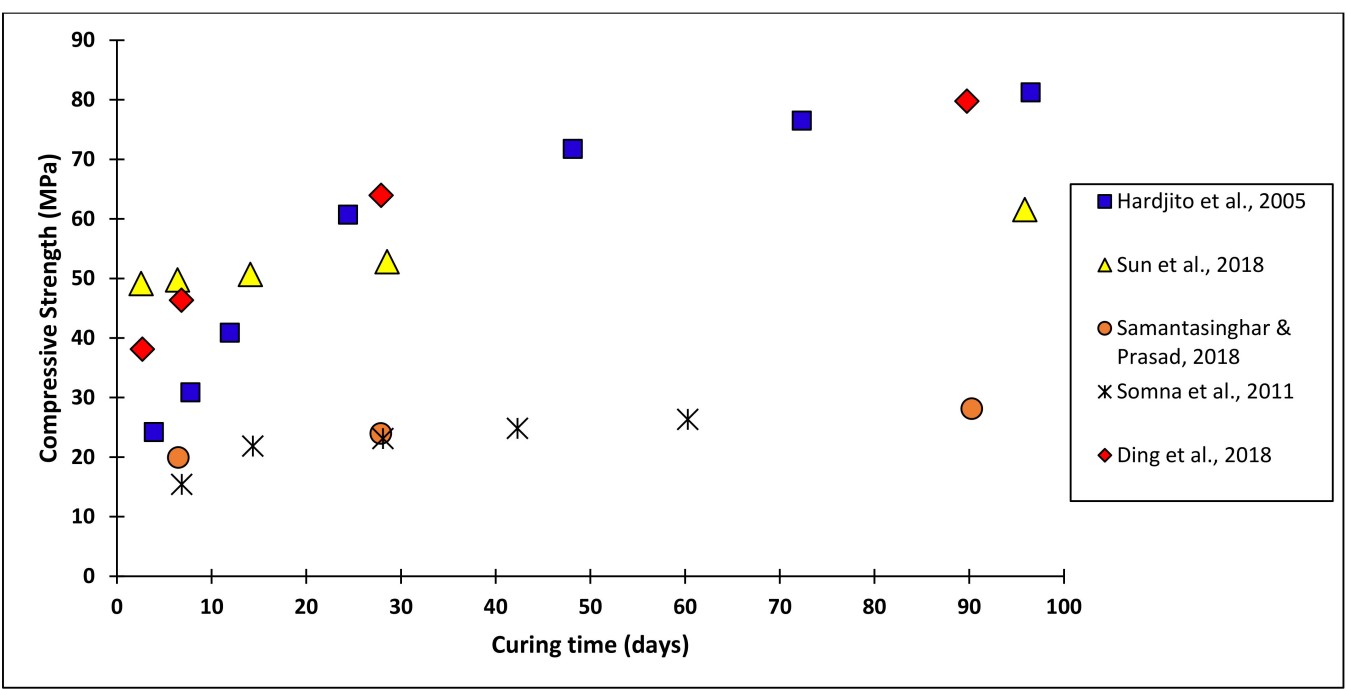

**Figure 4.** Graph compiling compressive strength versus curing time data obtained in previous investigations [54,92,97,101,161].

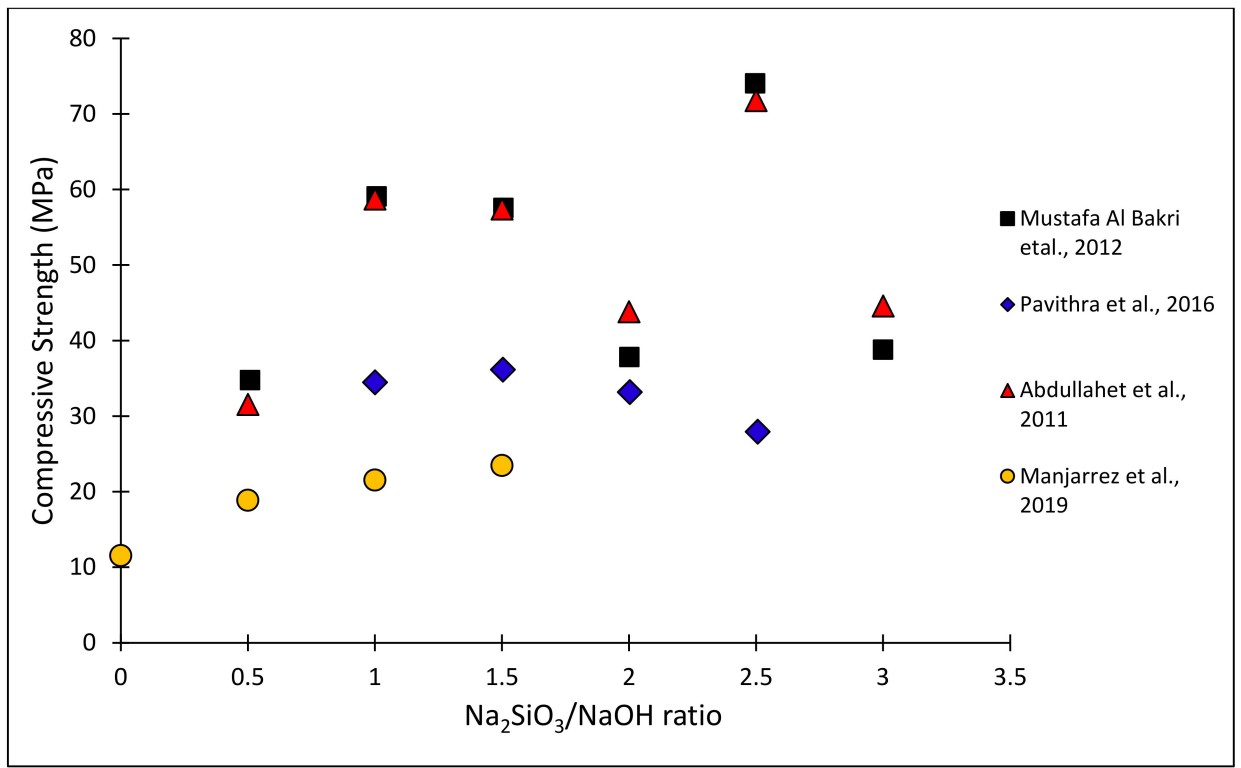

**Figure 5.** Graph compiling compressive strength data versus $Na_2SiO_3/NaOH$ ratio obtained in previous investigations [94,178,179,184].

Aida Mohd Mortar et al. [185] analyzed the behavior of fly ash and aggregate-based geopolymers with different molarities and amounts of the alkaline activator. The amount of alkaline activator was quantified by the binder/aggregate ratio. From the results it was obtained that the compressive strength increases with the molarity of the activator up to 14 M, higher concentrations of the activator produced a decrease in the compressive strength. In the same way, the compressive strength increases as the percentage of alkaline activator increases, the optimum being found at 35% activator and 65% aggregates, obtaining the best strength in all the concentrations analyzed.

A study by Somna et al. [54] where the effect of fly ash grinding prior to geopolymer formation was analyzed shows that increasing the concentration of the alkaline activator (NaOH) up to 14 M also increases the compressive strength of geopolymers for both unground fly ash (OFA) and ground fly ash (GFA) geopolymers. From this study it is also observed that pre-grinding the fly ash has a greater effect on the compressive strength of geopolymers than an increase in the concentration of the alkaline activator, mainly because there is a greater degree of release of the particles with the elements of interest for the formation of the solidifying gel.

A study by Ishwarya et al. [180] showed that an increase in the amount of alkaline activator in fly ash and slag based geopolymers produced an increase in the simple compressive strength. The optimum was found to be 30% by weight of alkaline activator in the total mixture. It should be noted that the samples cured for seven days showed no increase in compressive strength as the percentage of alkaline activator increased above 22%. From this it can be deduced that the geopolymerization process is a slow process and requires long curing times to obtain a good compressive strength, especially when working with a high percentage of alkaline activator.

As previously mentioned, there are many types of activators to carry out the geopolymerization process, so the final product to be obtained, which is the geopolymer, will also depend on the type of activator to be used. Although the activators derived from sodium are the most common, lately activators based on other elements have been used where sodium is not so effective as an alkaline activator (Table 6). It should be noted that the use of wastes from other industries as alkaline activators for the production of geopolymers is currently being influenced, as is the case of calcium carbide waste in the production of acetylene gas.

**Table 6.** Summary of activators used for the formation of geopolymers.

| Aluminosilicate Source | Activator | Reference |
|---|---|---|
| Copper tailings and fly ash | Sodium hydroxide | Zhang et al., [17] |
| Fly ash | Sodium silicate and sodium hydroxide | Burduhos Nergis et al., [41] |
| Garnet tailings and metakaolin | Sodium silicate | Wang et al., [70] |
| Fly ash and ground granulated blast furnace slag | Sodium carbonate and sodium silicate | Ishwarya et al., [180] |
| Gold mine tailings | Sugar mill lime sludge (Ca-based activator) | Opiso et al., [186] |
| Metakaolin and commercial furnace slag | Potassium silicate | Panizza et al., [126] |
| Fly ash | Calcium carbide residue and sodium silicate | Phetchuay et al., [187] |

*2.4. Water Content*

Water content in geopolymeric mixtures is an important parameter to consider, since higher water contents produce more fluid and less viscous pastes, which favors their preparation process, but at the same time has repercussions on the compressive strength of the geopolymers once hardened.

Xie and Kayali [188] conducted a study analyzing the behavior of fly ash-based geopolymers with different water contents, cured with heat and at ambient temperature. The study showed that for both types of curing, lower water contents in the geopolymeric blends resulted in more compact structures and higher compressive strength development. In addition, it was observed that lower initial water contents resulted in a higher rate of compressive strength gain in geopolymers cured at ambient temperature, whereas with heat cured geopolymers, lower initial water content has no effect on the rate of compressive strength gain.

It has been shown that the initial water content in geopolymer blends does not have much effect on the density of the pastes, but the density does have an important effect on the compressive strength [189].

A study by Khale and Chaudhary [23] shows that compressive strength decreases as the ratio of water-to-geopolymer solid by mass increases (Figure 6). This trend is analogous to water-to-cement ratio in the compressive strength in OPC. Although chemical processes involved in the formation of binders of both are entirely different [190]. The minimum water to cement ratio is approximately 0.4 by weight for Portland cement, whereas the fresh geopolymeric material is readily workable even at low liquid/solid ratio [191].

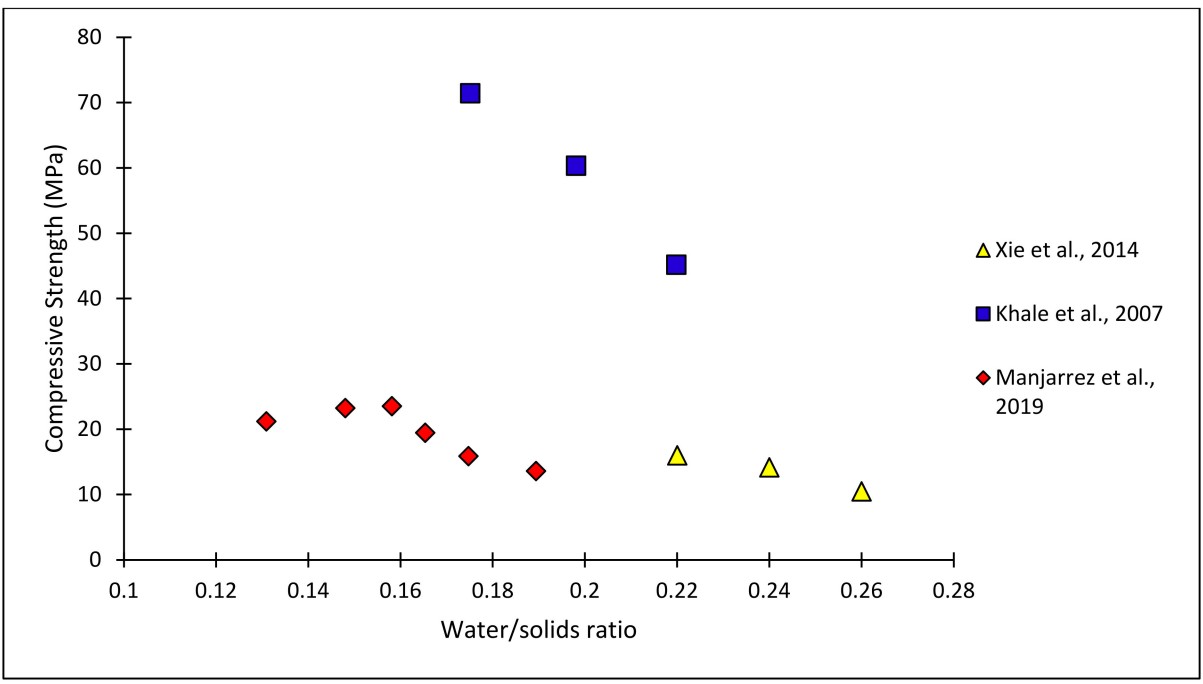

**Figure 6.** Graph compiling compressive strength data versus water/solids ratio obtained in previous investigations [23,94,188].

It has been shown that the initial water content in the formation of geopolymers is a fundamental factor in relation to the porosity of the product. Because higher water contents produce more porous geopolymers. Some authors relate this to the evaporation of free water, which produces porosity and residual stress [192,193].

In a study conducted by Manjarrez et al. [94], the effect of water content in geopolymers based on copper tailings and low calcium slag was analyzed. The results showed that in specimens cured for seven days at 60 °C, the compressive strength increased slightly with water content and then declined (Figure 6). According to the authors, this is due to the fact that with small amounts of water, lumps were formed in the mixture and large voids were formed in the specimens. When the water content increased, a maximum compressive strength of 23.5 MPa with a w/s ratio of 0.158 was obtained. Further increases in water content led to lower compressive strengths due to reduced particle–particle interaction. According to other authors, this is due to the fact that a higher amount of water causes a lower degree of polycondensation [194].

From the data collected on strength versus water/solids ratio, it can be generally stated that higher water contents in geopolymer blends lead to lower compressive strengths (Figure 6); however, very low water contents can also lead to losses in the compressive strength of geopolymers as previously mentioned. It is worth mentioning that the water content to be used also depends on the source of aluminosilicates, and on the alkaline solution used, since different compositions generate mixtures with different fluidities, so the water content to be added also varies. In addition, a smaller amount of water implies using less amount of the activating reagent to reach the desired concentration, so

optimizing the amount of water to be used is an important parameter to consider. Since the water contents used in geopolymeric mixtures are generally low, the pastes produced have high yield stresses and viscosities, so that their eventual transport and molding could be difficult.

### 2.5. Effect of Presence of Calcium and Other Impurities

Yip et al. [195] conducted an investigation and in it they observed the influence of calcium sources on geopolymerization. He used seven sources, two corresponding to processed sources and the rest were natural crystalline calcium sources. These sources were mixed with metakaolin in different proportions and sodium silicate, the latter at different ratios adjusted with NaOH (R). From these mixtures, the results of Table 7 were obtained, where each mixture was differentiated by label in the matrix column.

**Table 7.** Compressive strength with different amounts of calcium and alkaline solution [195].

| Calcium Silicate Material (CS) | MK/MK + CS | R = 2.0 | | | R = 1.5 | | | R = 1.5 | | |
|---|---|---|---|---|---|---|---|---|---|---|
| | | Matrix | 7-Day [MPa] | 28-Day [MPa] | Matrix | 7-Day [MPa] | 28-Day [MPa] | Matrix | 28-Day [MPa] | 7-Day [MPa] |
| None | 1 | S1 | 34.6 | 35.2 | S2 | 62 | 65 | S3 | 36.2 | 38.4 |
| GGBFS | 0.8 | A1 | 47.1 | 54.2 | A3 | 45.3 | 46.8 | A5 | 38.6 | 40.5 |
| | 0.6 | A2 | 41.5 | 52.7 | A4 | 38.6 | 39.3 | A6 | 25.4 | 26.0 |
| CEM | 0.8 | B1 | 47.5 | 53.5 | B3 | 49.3 | 56.8 | B5 | 46.2 | 51.4 |
| | 0.6 | B2 | 31.2 | 28.1 | B4 | 35.4 | 35.1 | B6 | 32.2 | 33.8 |
| WOL | 0.8 | C1 | <5.0 | 18.8 | C3 | 36.5 | 38.2 | C5 | 22.7 | 25.3 |
| | 0.6 | C2 | <5.0 | 16.8 | C4 | 19.3 | 24.3 | C6 | 14 | 20.8 |
| HRN | 0.8 | E1 | <5.0 | 8.3 | E3 | 31.1 | 36.7 | E5 | 32.3 | 37 |
| | 0.6 | E2 | <5.0 | 5.7 | E4 | 21.3 | 23.3 | E6 | 17.3 | 22.4 |
| TRM | 0.8 | G1 | N/A | N/A | G3 | 31.7 | 38.3 | G5 | 27.8 | 35.3 |
| | 0.6 | G2 | N/A | N/A | G4 | 26.5 | 28.6 | G6 | 19.5 | 25.4 |
| PRH | 0.8 | F1 | 6.7 | 14.3 | F3 | 32.2 | 39.4 | F5 | 29.5 | 36.4 |
| | 0.6 | F2 | 6.2 | 11.5 | F4 | 24 | 25.1 | F6 | 14 | 21.2 |
| ANO | 0.8 | D1 | N/A | N/A | D3 | 29.3 | 35.3 | D5 | 26.2 | 28.7 |
| | 0.6 | D2 | N/A | N/A | D4 | 18.8 | 22.1 | D6 | 15.8 | 20.3 |

Note: MK (metakaolin); GGBFS (ground granulated blast furnace slag); CEM (cement); WOL (wollastonite); HRN (hornblende); TRM (tremolite); PRH (prehnite); ANO (anorthite).

The results show that the dissolution of calcium in sources processed in a low alkalinity system provides good compressive strength due to the formation of CSH (Calcium Silicate Hydrate) gel [196] that coexists with the geopolymer gel (NASH), which subsequently complement each other and form a mostly amorphous structure that provides a better mechanical behavior. Very little calcium dissolves from natural sources, thus preventing the formation of CSH gel to form and causing the calcium crystals to interrupt the geopolymer gel. This indicates that the crystal disrupts the amorphous gel structure and stiffens the material, subsequently generating cracks around it, and reducing the compressive strength.

In high alkalinity systems (higher concentration of NaOH), the formation of NASH gel is predominant, where the role of calcium is less influential in the final product as it cannot generate a CSH gel that contributes to the geopolymer gel or a large number of crystals that disrupts the amorphous structure of the gel. Therefore, the dissolution of calcium does not have a decisive impact on the compressive strength [195].

Yip et al. [195] focused on replacing part of the aluminosilicate source with calcium. Similarly, Tian et al. [197] studied the effect of replacing part of the alkaline activator (NaOH) with calcium oxide (CaO) produced industrially to form a geopolymer based on copper tailings from China with the same composition of their work when varying the temperature cured [81] and fly ash. It was observed that replacing 20% of the alkaline activator with CaO provides a greater compressive strength; however, this slightly reduces the compressive strength in the long term (28 days of aging at ambient conditions) since the best strength was obtained within seven days. Moreover, by means of micrographs, it

was observed that Si, Al, and Na accumulate in a zone that would correspond to the NASH gel and the zone with absence of these elements where the Ca concentration predominates implies that there was no interaction of Ca with the rest of the elements to form a CSH gel.

Pachecho-Torgal and Jalali [198] studied how the presence of $Ca(OH)_2$ influences the curing time at ambient conditions and the hydroxide concentration to obtain different compressive strengths when using tungsten tailings as a source of aluminosilicates (considering that they performed a heat treatment prior to tailing at 950 °C for 2 h to generate more soluble phases). In the study, a phenomenon similar to that described above was observed, where the higher the alkalinity of the system, the greater the gain in compressive strength, with this effect being more noticeable the longer the curing time. Regarding the presence of $Ca(OH)_2$, it was observed that the compressive strength achieves an optimum point when the calcium content rises to 10%, and higher amounts of calcium hydroxide subsequently led to a loss of compressive strength.

A study conducted by Yousefi Oderji et al. [199] in which they studied the behavior of fly ash based geopolymers with different amounts of high calcium slag, shows that an increase in the slag content leads to an increase in the compressive strength of the geopolymers, due to the high CaO content in the slag [162], the optimum point being a 15% replacement of fly ash by slag. The high CaO content in the fly ash can increase the compressive strength due to additional hydration reaction that may take place [200]. This can be attributed to the formation of C-A-S-H gels, which would reduce the porosity and condense the microstructure of alkali activated geopolymer matrix [113,201–204].

Fly ash and slags often contain considerable calcium, in which case coexistence of geopolymer gel and calcium aluminosilicate hydrate (C-A-S-H) is often observed [205]. Additionally, CaO content in the fly ash have significant effect on the setting time of geopolymer concrete [206].

Komnitsas et al. [207] studied the behavior of a geopolymer synthesized from an iron-rich slag activated with KOH, to which he added different types of nitrates and sulfates in different amounts. From the results obtained, it was observed that, except for the case of lead nitrates and sulfates, which decrease the compressive strength more slowly, the compressive strength drops drastically when adding nitrates and sulfates of any type, highlighting that having a 1% $w/w$ of chromium nitrate, and the geopolymer was no longer formed. The behavior of copper nitrates and sulfates are like each other along with nickel nitrate. Looking for heavy metals, it can be determined that nitrates have a more negative behavior than sulfates. This can be explained by the fact that nitrates and sulfates rapidly consume the moles of KOH, especially chromium, which its nitrate variant consumes between 80 and 100% of potassium hydroxide. The rest of the nitrates and sulfates can consume around 20% to 100% of the moles of the alkaline solution, except for lead, which in its nitrate variant managed to consume up to 60% of KOH. This prevents enough moles of the alkaline solution to dissolve the silicon and aluminum from the source and the geopolymer to form.

In summary, Table 8 presents the data collected on impurities and how they were presented, also how their influence and the effect they had on compressive strength were evaluated. It can be mentioned that the addition of calcium in low amounts (not exceeding 20% by weight) are favorable for the production of geopolymers because they increase the compressive strength of these, due to the effect they have on the formation of gels that coexist with the NASH geopolymer gel. This calcium can come from the aluminosilicate source as well as from the alkaline activator, being the effect produced by this calcium dependent on the source from which it is obtained.

**Table 8.** Summary on impurities in the formation of geopolymers.

| Source | Impurity | Type | Experiment | Effect | Optimal Point | UCS (MPa) | Reference |
|--------|----------|------|------------|--------|---------------|-----------|-----------|
| Metakaolin | Calcium | Processing (slag and cement) and natural crystals | Aluminosilicate source replacement | Processed: positive to low alkalinity. Natural crystals: negative at low alkalinity. At high alkalinity the effect is null for both types. | 20% (cement) | 56.8 | Yip et al. [195]. |
| Tailings | Calcium | Calcium oxide | Alkaline solution replacement | Positive at optimal point | 20% | ~40 | Tian et al. [197]. |
| Tailings | Calcium | Calcium hydroxide | Presence in the source of aluminosilicates | Positive at optimal point | 10% | 70 | Pacheco-Torgal and Jalali [198] |
| Iron slag | Lead, nickel, copper, and chromium | Sulfates and nitrates | Added in the mix for the formation of the geopolymer | Negative | - | - | Komnitsas et al. [207]. |

It can be assumed that the presence of calcium can have positive effects on the formation of the geopolymer if it can be controlled in its proportion, either in the source of aluminosilicates or in the alkaline solution. Given that calcium is present in the source of aluminosilicates, it would only have an effect on the formation if it is presented as calcium hydroxide, as we are aware calcium from processed sources and as natural crystals will have no effect on the compressive strength, as the formation of geopolymers occurs in a highly alkaline system. Finally, the presence of heavy metals such as lead, nickel, copper, and chromium have negative effects on the compressive strength, so their content should be analyzed when choosing the source of aluminosilicates that we would prefer to use.

Previous studies have proven the ability of geopolymers to encapsulate toxic elements from industrial wastes, so that these wastes can be reused and are not hazardous to the environment. By means of leachability tests, it has been obtained that geopolymers encapsulate elements such as Pb, Cu, Zn, and As, whereby using cementation methods, the leachability of industrial wastes containing these elements has been drastically reduced [208,209].

## 3. Conclusions

From the findings of the previous studies that investigated the mechanical properties of geopolymers. The conclusions are summarized as follows:

It has been shown that the compressive strength of geopolymers depends mainly on their chemical composition, a fundamental parameter to be considered being the source of aluminosilicates. Elements present in the source such as silicon and aluminum are fundamental for the formation of the N-A-S-H gel, which is responsible for the hardening of geopolymers. This is why the ratio between these elements is an important parameter to consider in the production of geopolymers, in order to obtain good compressive strengths. On the other hand, it has been demonstrated that the different aluminosilicate sources have different reactivities against alkaline agents, mainly due to their origin. Sources with higher reactivity have been shown to produce geopolymers with higher strengths.

The geopolymeric reactivity of aluminosilicate materials can be significantly improved by using several techniques. These activation processes enhance reactivity by increasing the rate and extent to which silicon and aluminum species from the activated materials dissolve in alkaline medium [210].

It is evident that temperature and time play a fundamental role in the curing of geopolymers, showing that higher temperatures and curing times promote the formation of the N-A-S-H gel and consequently the hardening and strength gain of the geopolymers. However, this depends on the composition of the geopolymer, since it has been demonstrated in several studies that geopolymers can be cured at ambient temperature without the need to apply external heat. Although applying heat favors the geopolymerization reactions, it is not necessary in all cases because it can be complemented by applying longer curing times to geopolymers cured at ambient temperature.

With respect to alkaline activators, in most cases sodium derivatives were used. Some studies used a 100% sodium hydroxide solution, while others used a mixture of sodium hydroxide and sodium silicate, because it has been shown that the latter results in geopolymers with better microstructures, denser matrices, and, consequently, improved compressive strength. In addition, most studies show that an increase in the amount of alkaline activator causes an increase in compressive strength.

Water content is an important parameter to consider in the production of geopolymers. It has been observed that lower amounts of water produce geopolymers with better compressive strengths, and lower water use means less use of alkaline reagents. On the other hand, a higher water content produces geopolymer pastes of higher fluidity but obtaining a lower resistance in the already hardened geopolymers.

Small amounts of calcium have proven to be beneficial for obtaining good compressive strength due to the formation of the C-A-S-H gel that coexists with the N-A-S-H gel, while high amounts of calcium have proven to be detrimental to the compressive strength of geopolymers. This happens too in the presence of heavy metals such as copper, chromium, and nickel with a negative influence in the compressive strength.

**Author Contributions:** H.C. (Hengels Castillo): Writing—review & editing, Principal investigator, Project administration, Resources, Supervision, Validation; H.C. (Humberto Collado): Writing—review & editing, Investigation; T.D.: Writing—review & editing, Investigation; S.S.: Writing—review & editing, Investigation; M.V.: Resources, Supervision, Validation; P.G.: Resources, Supervision, Project administration; S.P.: Writing—review & editing, Investigation, Validation, Supervision. All authors have read and agreed to the published version of the manuscript.

**Funding:** This research received no external funding.

**Acknowledgments:** We are grateful for the support of this research project to the Consulting Company JRI Engineering from Chile (https://www.jri.cl/, accessed on 16 November 2021) and the Center for Research in Sustainable Mining CIMS-JRI (https://www.cimsjri.cl/, accessed on 16 November 2021).

**Conflicts of Interest:** Authors state no conflict of interest.

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
