# Peer review of "Factors Affecting the Compressive Strength of Geopolymers: A Review"

_minerals, doi:10.3390/min11121317_

Round 1

Reviewer 1 Report

1. The abstract needs to be significantly expanded.
2. Review article of 37 references is rather scarce. I agree that all the classic works on geopolymers are presented. But such reviews have already been done more than once. I suggest adding a comparison with other types of concrete. For example:
- Elistratkin, M.Y., Lesovik, V.S., Zagorodnjuk, L. H. New point of view on materials development. IOP Conference Series Materials Science and Engineering 2018. 327(3), 032020     DOI:10.1088/1757-899X/327/3/032020 
- Volodchenko, A.A., Lesovik, V.S., Cherepanova, I.A. Peculiarities of non-autoclaved lime wall materials production using clays. International conference on mechanical engineering, automation and control systems 2017. 2018. 327
- G. Murali , Roman Fediuk. A Taguchi approach for study on impact response of ultra-high-performance polypropylene fibrous cementitious composite. Journal of Building Engineering 30 (2020) 101301. https://doi.org/10.1016/j.jobe.2020.101301
- Semenov, PA; Uzunian, AV; Davidenko, AM et al. First study of radiation hardness of lead tungstate crystals at low temperatures. Nuclear instruments & methods in physics research section A: Accelerators spectrometers detectors and associated equipment. 2007. 582(2). pp. 575-580. DOI: 10.1016/j.nima.2007.08.178

3. The goal should be defined more clearly from the standpoint of scientific novelty.
4. Why in table 2 marking Na52 corresponds to 50% SODIUM? Moreover, in Figure 2-5, this sample is no longer there.
5. By the way, the numbering of figures must be single-digit
6. Figures in reviews should not be taken from one ref, but made by the author independently during the analysis of various articles

Reviewer 2 Report

Reviewer comments:

The review paper includes information on the Effect of different parameters on the Mechanical Properties of Geopolymer Composites. The text is a very good technical report, which would be useful in terms of Geopolymer Technology.

The manuscript lacks major important information. These are listed below:

1) The different phases of the geopolymer binder should highlight in the SEM image – Figures 2-15, 2-16, 2-32 and 2-33(a).

2)  Authors have cited very old references e.g. 2003, 2004, 2005, 2006, 2007, 2009 please include current references.

3) To produce a very impactful review paper, an author should refer to more than 200 research articles. The authors have included very few studies. 37 references are very few for one good review paper. The author should add more findings from the previous studies. Delete some irrelevant figures.

4) Author should revise the introduction and conclusion part.

5) The author should have to change the manuscript title.

Reviewer 3 Report

The abstract is too weak with an unclear objective of the study is presenting. The manuscript is hampered by several syntax and grammatical errors. There is no strong justification or critical review in this manuscript. The title is too general while the manuscripts are focused on the factors that affected the geopolymer properties. The material selection should be selected for several aluminosilicate materials to gain knowledge attraction.   

The following suggestion and comments should be taken

  • Abstract. “…tendency to obtain high compressive strength with Si/Al ratios around 2, together with a curing temperature of 90ºC.” Geopolymer can be produced by various aluminosilicate materials. Different materials consist of different chemical compositions. Hence, Si/Al will be obtained at the different values to produce higher strength. Also, not all geopolymers are required for oven curing. 
  • Please check the manuscript format. Standardize font type and size throughout the manuscript.
  • “…the effect of the unreacted material present, which act as inhibitors in the binder.” The unreacted materials are due to the type of material used as a geopolymer binder which consists the various chemical composition. Thus, an impossible Si/Al ratio is involved. 
  • “…Si/Al = 2 ratios with a simple compressive strength of 36.8 MPa. The Na/Al and H2O/NaO ratios are similar to those used by Duxson et al.” Did Duxson used the similar material? It is impossible to compare the oxide molar ratio for the different materials.
  • What did the meaning of liquid silicon?
  • “…the resistance results presented by Subaer, where it’s evident the highest resistance is achieved when using a low Na/Al ratio.” This depends on Si/Al ratios. One Na+ is occupied with Al monomer, the balanced free Na will react with Si-O-Al chain to form NASH, hence obtain higher strength.
  • Figure 2-8. There is less discussion on lower porosity towards lower strength. This could be probably due to the other factor such as curing age.
  • The term of resistance of compression is confusing, please use “compressive strength” and standardize throughout the manuscript.
  •  Figure 2-11. Na2O/Si2 is preferable to be written as Na2O/SiO2.
  • “Higher amounts of silica led to a loss in compressive strength”. Please give a more critical review of this statement.
  • “…Si/Al ratios of 1:1 a small N-A-S-H gel forms…” How did the author determine the size of NASH?
  • The formation of pores is could be affected by the liquid component in the geopolymer system, to be more specific is the effect of solid/liquid ratios.
  • “…alumina-silicate and the leaching of ions from this slag is much higher under a given alkaline condition than from fly ash.” Commonly, slag are deliver high CaO which is believed to contribute to high strength and fast setting.
  • Calcium-Silicon-H2O is not referring to CSH. 
  • Table 2-7. The data is not significant to the relevant text. 
  • There is a less critical review on calcium contribution towards geopolymer behavior.

Round 2

Reviewer 1 Report

The manuscript is very crude and not ready for publication in this authoritative edition of the Q1 Web of Science. The title is completely illiterate and does not give any idea of the novelty that should be contained (but is not contained in this manuscript). The abstract has been thoughtlessly expanded after my previous comment, but so thoughtlessly that not a single numerical comparative result is given. But this is a technical article!

Author Response

Dear reviewer. We have improved the paper, its writing and content. The title has been adapted to the respective review. The resume has been improved, adding technical comparisons, respectively. Thanks.

Reviewer 2 Report

The author should check is there any need for copyright permission for figures i.e. SEM image figures.

Author Response

Dear reviewer, given the times that are always very limited, we have avoided placing images that require author's permissions, and therefore, in the particular case of your comment, we have eliminated the SEM images, and also, the graphics have been made by us in an original way as comparisons between different authors. Thanks.

Reviewer 3 Report

The authors made some revisions to the manuscript. However, the response is mentioned in detail which/what part has been changed in the revised manuscript. It is difficult for the reviewer to validate the changes that have been made by the author. 

Specific comments are provided as follows:

  • Response 1: Please use the term “Review” seem it was a Review Article
  • Response 1: Alkali activated and geopolymer is totally different things. Please revise wisely. 

Author Response

Dear Reviewer, we have identified with yellow the modifications that have been made since the original paper so that you can better identify the changes. The title has been modified. Figures and graphics were modified to avoid the rights and permissions of the respective authors. Thanks.

Round 3

Reviewer 1 Report

To be honest, the third version is not much improved. The way the authors make corrections, highlighting entire chapters in yellow, tells us that we are confused to fool our brains. I ask you to carefully look at my comments again and carefully answer each of them. While my opinion is unchanged!

Author Response

Dear Reviewer,

thanks for your comments.

Regarding your observations, as a team we believe that we have responded to each of your observations, which I will comment on again:

The abstract has been written better. The reviewer should consider that the study we are conducting is regarding geopolymers based on copper flotation tailings and we have sought references regarding these investigations, however, we have expanded as suggested by you.

According to your comments, more review papers and comparisons between them have been added, so that a more comprehensive review of the particular topic results. For this review, 200 papers are presented that have been considered, however, please, I ask the reviewer to take into consideration that review papers with 48 references have been accepted in the Minerals Journal (see "Lessons from tailings dam failures - where to go from here? "by the author" David Jhon Williams ").

The abstract and the introduction have been modified and expanded for a better understanding according to the reviewer's considerations.

The colors yellow, and now green, are due to the fact that the other reviewers have requested to mark what the modifications have been between one version and another. For this new version, the modified and improved has been marked with green.

Thank you for allowing us to continue improving our paper.